# The Impact of the Use of Glycomacropeptide on Satiety and Dietary Intake in Phenylketonuria

**DOI:** 10.3390/nu12092704

**Published:** 2020-09-04

**Authors:** Anne Daly, Sharon Evans, Alex Pinto, Richard Jackson, Catherine Ashmore, Júlio César Rocha, Anita MacDonald

**Affiliations:** 1Dietetic Department, Birmingham Children’s Hospital, Steelhouse Lane, Birmingham B4 6NH, UK; evanss21@me.com (S.E.); alex.pinto@nhs.net (A.P.); catherine.ashmore@nhs.net (C.A.); anita.macdonald@nhs.net (A.M.); 2Liverpool Clinical Trials Centre, University of Liverpool, Brownlow Hill, Liverpool L69 3GL, UK; r.j.jackson@liverpool.ac.uk; 3Nutrition and Metabolism, NOVA Medical School, Faculdade de Ciências Médicas, Universidade Nova de Lisboa, 1169–056 Lisboa, Portugal; rochajc@nms.unl.pt; 4Centre for Health and Technology and Services Research (CINTESIS), 4200–450 Porto, Portugal

**Keywords:** phenylketonuria, PKU, glycomacropeptide, satiety

## Abstract

Protein is the most satiating macronutrient, increasing secretion of gastrointestinal hormones and diet induced thermogenesis. In phenylketonuria (PKU), natural protein is restricted with approximately 80% of intake supplied by a synthetic protein source, which may alter satiety response. Casein glycomacropeptide (CGMP-AA), a carbohydrate containing peptide and alternative protein substitute to amino acids (AA), may enhance satiety mediated by its bioactive properties. Aim: In a three-year longitudinal; prospective study, the effect of AA and two different amounts of CGMP-AA (CGMP-AA only (CGMP100) and a combination of CGMP-AA and AA (CGMP50) on satiety, weight and body mass index (BMI) were compared. Methods: 48 children with PKU completed the study. Median ages of children were: CGMP100; (*n* = 13), 9.2 years; CGMP50; (*n* = 16), 7.3 years; and AA (*n* = 19), 11.1 years. Semi-quantitative dietary assessments and anthropometry (weight, height and BMI) were measured every three months. Results: The macronutrient contribution to total energy intake from protein, carbohydrate and fat was similar across the groups. Adjusting for age and gender, no differences in energy intake, weight, BMI, incidence of overweight or obesity was apparent between the groups. Conclusion: In this three-year longitudinal study, there was no indication to support a relationship between CGMP and satiety, as evidenced by decreased energy intake, thereby preventing overweight or obesity. Satiety is a complex multi-system process that is not fully understood.

## 1. Introduction

Phenylketonuria (PKU), due to phenylalanine hydroxylase deficiency, leads to accumulation of phenylalanine and irreversible brain damage if untreated [1,2]. A lifelong phenylalanine/protein restricted diet is essential, and most patients with classical PKU tolerate ≤ 500 mg/day of phenylalanine (equivalent to ≤10 g/day protein). Meats, fish, eggs and cheese are avoided with foods such as potatoes, cereals and peas given in restricted and calculated amounts; special low protein foods together with some fruits and vegetables (containing phenylalanine ≤ 75 mg/100 g) are given without restriction. Therefore, the diet is lacking in high quality protein and protein intake is supplemented with a minimal/phenylalanine-free protein substitute, usually supplying up to 80% of protein requirements. It is unclear if this synthetic source of protein alters satiety. Satiety, which is a sense of fullness after eating, is important in regulating food intake [3,4]. In general nutrition, there is evidence that the amount and type of dietary protein alters appetite and may influence weight regulation [5,6].

Protein substitutes for PKU are obtained from either casein glycomacropeptide (CGMP) or phenylalanine-free amino acids (AAs). CGMP and AAs are compositionally different. Amino acid-based protein substitutes are composed of free L-amino acids only, whereas CGMP is a glycosylated peptide containing varying amounts of oligosaccharides, mostly sialic acid (N-acetylneuraminic acid), galactosamine and galactose [7]. CGMP has prebiotic, antimicrobial and immunomodulatory effects [8,9] and is prevalent in bovine milk. It constitutes 20–25% of the total protein in whey products [10]. Whey protein has been shown to induce satiety to a greater extent than other protein sources such as casein, soya and egg albumin [11,12]. There is some evidence that CGMP influences hormone responses affecting satiety [13,14,15]. However, human studies investigating the effect of CGMP on food intake and satiety have resulted in mixed findings [16,17,18,19].

Gut absorption of protein is thought to modulate satiety, although the influence of the protein source and individual amino acids on the control of food intake is not completely understood. It involves complex pathways that affect vagus-mediated signals, satiety related hormones and their metabolites (including the peptide ghrelin, cholecystokinin (CCK), glucagon-like peptide-1 (GLP-1), glucose dependent insulinotropic polypeptide (GIP) and peptide tyrosine–tyrosine (PYY)) [11,15,20,21,22,23]. Some blood amino acids, particularly leucine, lysine, tryptophan, isoleucine and threonine, are linked to satiety responses [24,25,26,27]. CGMP is a small peptide and likely to be quickly absorbed [22,28,29]. A sharp rise in the mean circulating amino-acid concentrations following protein ingestion has been associated with a reduction in appetite [30,31]. Korompokis et al. [32] investigated the absorption kinetics of protein and the impact of amino acids on appetite and satiety related hormones. In a randomized cross over study using liquid preloads with a similar energy density but variable energy from protein, carbohydrate and fat, no postprandial kinetic effect of amino acids on appetite was shown. They concluded protein intake affected the amino acid profile but was not related to appetite regulation. 

Equally, it is well established that amino acids bypass degradation by proteases and compared to whole proteins are absorbed faster [33,34,35,36]. In our own center, we reviewed plasma amino acid concentrations both fasting and after 2 h following a breakfast meal and 20 g protein equivalent from protein substitute either based on AA or CGMP. Although there were significant differences for individual amino acids, probably related to the amino acid composition of the protein substitutes, postprandial total amino acid concentrations were not different between the protein substitutes [37]. Until the absorption kinetics of CGMP (CGMP with added rate limiting amino acids) has been reported, its influence on satiety remains speculative. Additionally, CGMP and AA supplements modified for PKU usually contain added carbohydrate, which may also influence amino acid kinetics and hormone response [28].

If CGMP did increase satiety in PKU as suggested [38], it may bring important benefits, potentially helping control obesity, commonly reported in PKU [38,39]. In this study, we compared energy intake, weight and body mass index (BMI) over a three-year period in a group of children with PKU taking either CGMP at two different concentrations or amino acid supplements. We hypothesized that, if CGMP influenced satiety, then energy intake should be lower and weight and BMI changes altered between the groups.

### Ethical Permission

The South Birmingham Research Ethics committee granted a favorable ethical opinion, referenced 13/WM/0435 and IRAS (integrated research application system) number 129497. Written informed consent was obtained for all subjects from at least one caregiver with parental responsibility and written assent obtained from the subject if appropriate for their age and level of understanding.

## 2. Methods and Materials

### 2.1. Subjects

In a three-year long-term prospective study, 50 children (28 boys, 22 girls) with PKU were recruited. Their median recruitment age was 9.2 years (range 5–16 years). Forty-seven children were European and three were of Pakistani origin. Inclusion criteria included: diagnosed by newborn screening, aged 5–16 years, not treated with sapropterin dihydrochloride and adherent to protein substitute. Seventy percent of routine blood phenylalanine concentrations were within phenylalanine target range for six months before study enrolment. Target blood phenylalanine concentrations for children aged 5–11 years was <360 µmol/L and for 12 years and older <600 µmol/L as recommended by the European PKU guidelines [40]. 

### 2.2. Protein Substitutes

Two types of protein substitute were studied: one based on phenylalanine-free AA (liquid pouches or powders) and the other on a powdered CGMP-AA supplement (a study product, made by Vitaflo, International Ltd, UK) (see Table 1 for nutritional analysis comparisons). The CGMP contained a residual amount of phenylalanine (36 mg/20 g protein equivalent). It was supplemented with essential and semi-essential amino acids, to provide a balanced amino acid profile to sustain nitrogen requirements and so the term CGMP-AA is used. 

### 2.3. Study Design

The primary aim of this three-year single center, longitudinal study was to compare the efficacy of CGMP-AA compared to AA on bone health in a group of 50 PKU children (this will be reported separately). The secondary aim was to study energy intake, with particular reference to protein intake (from protein substitutes and phenylalanine exchanges), weight and BMI changes between the two groups, exploring the theory that CGMP-AA is actively related to satiety. Following the findings from a pilot study [41], it was clear that not all children in the CGMP-AA group were able to tolerate their entire protein substitute from CGMP-AA, due to its phenylalanine content. Therefore, within this group, there was a further subdivision: CGMP100, those taking all their substitute from CGMP-AA and those taking a combination of CGMP-AA and AA, named CGMP50. A group of children remained on AA only.

### 2.4. Selection into AA or CGMP-AA Group

The children chose AA or CGMP-AA, depending on their taste preference. They remained on this protein substitute for the three-year duration of the study.

### 2.5. Dietary Assessment

A three-day semi-quantitative dietary assessment was completed once every three months. The diet diaries were all checked during a face-to-face interview with caregivers by one of two trained metabolic dietitians. Protein containing foods were weighed. A picture book of pre-weighed foods was used to help caregivers estimate the portion size of other foods such as low protein pasta, fruit and vegetables. Twice a year, average food portions were weighed, e.g., low protein pasta, bread and low protein sausages. All children were observed eating at least one meal annually. 

### 2.6. Nutritional Analysis

The dietary assessments were analyzed using Nutritics Nutritional Software (v5.093). The following macronutrients were analyzed: daily energy (Kcal), protein (g), carbohydrate (unrefined and refined) (g) and fat (g). For each subject, the annual median macronutrient value was calculated, and the median value for all subjects in each group was determined, giving the median of the median value for each macronutrient. The results were compared with age and gender specific UK dietary reference values or estimated average requirement (EAR) for energy (UK Scientific Advisory Committee on Nutrition (SCAN)) [42]. The nutrient contribution from CGMP-AA and AA were included in this analysis as well as all special low protein foods. 

### 2.7. Anthropometric Measurements

Weight, height and BMI were measured once every three months by one of two metabolic dietitians. Height was measured with a Harpenden stadiometer (Holtain Ltd, Crymych, UK) and weight on calibrated digital scales (Seca UK model 875); they were measured to the nearest 0.1 cm and 0.1 kg, respectively.

### 2.8. Blood Phenylalanine Levels

Trained parents/caregivers collected weekly early morning fasted blood spots on filter cards, Perkin Elmer 226 (UK Standard NBS). Blood specimens were sent via first class post to the laboratory at Birmingham Children’s Hospital. All the cards had a standard thickness and the blood phenylalanine concentrations were calculated on a 3.2-mm punch by MS/MS tandem mass spectrometry.

### 2.9. Statistical Analysis

Continuous data are represented as median (IQR) and categorical data were summarized as frequencies of counts and associated percentages. Analyses of study endpoints were performed using Analysis of Covariance (ANCOVA) techniques, which analyzes the data at three years follow-up while adjusting for baseline values. As there was a difference in age between participants between the two groups, all models were adjusted for patient age as well as gender. The impact of CGMP compared to AA was evaluated by comparison of CGMP100 and CGMP50. Descriptive statistics are reported as medians and differences at baseline and follow up were assessed using a paired t test. Analysis was performed using R (Version 3).

## 3. Results

Of the 50 children recruited, 48 completed the study: CGMP group, *n* = 29; AA group, *n* = 19. The CGMP group was divided into CGMP100, *n* = 13 (45%), and CGMP50, *n* = 16, (55%). The median ages at enrolment were: CGMP100, 9.2 years; CGMP50, 7.3 years; and AA, 11.1 years. There was a significant difference in age between the AA and CGMP50 (*p* = 0.005) and between CGMP50 and CGMP100 (*p* = 0.04).

Prior to starting the CGMP-AA, all patients were prescribed AA as their source of protein substitute. Six subjects took powdered amino acids (XP Maxamum (Nutricia Ltd., Trowbridge, UK), *n* = 1; PKU Anamix first spoon (Nutricia Ltd.), *n* = 3; PKU gel (Vitaflo International Ltd.), *n* = 2) and 44 subjects took liquid pouches (PKU Lophlex LQ (Nutricia Ltd.), *n* = 3; PKU Cooler (Vitaflo International Ltd., Liverpool, UK), *n* = 41). The AA group remained on their usual protein substitute (PKU Lophlex LQ (Nutricia Ltd.), *n* = 1; PKU Cooler (Vitaflo International Ltd) *n* = 14) or a powdered preparation (PKU gel (Vitaflo International Ltd.), *n* = 4) throughout the study. The median (range) phenylalanine concentrations at baseline were not statistically different: CGMP100, 255 μmol/L (170–360); CGMP50, 290 μmol/L (220–430); and AA, 315 μmol/L (215–600). The majority had classical PKU, except two children who were mild based on untreated blood phenylalanine levels at diagnosis and dietary phenylalanine tolerance.

The median daily dose of protein equivalent from protein substitute was 60 g/day (range 40–80 g), and the median amount of prescribed natural protein was 5.5 g protein/day in all groups (range 3–30 g) or 275 mg/day phenylalanine (range 150–1500 mg).

### 3.1. Subject Withdrawal

One boy and one girl (aged 12 and 11 years, respectively) in the CGMP-AA group were excluded from the study, as both were unable to adhere with the study protocol. One failed to return blood phenylalanine samples and both had poor adherence to their phenylalanine restricted diet.

### 3.2. Nutritional Intake

#### Change in Nutrient Intake

In all the groups, the energy intake expressed as a percentage of EAR decreased over the three years. In the AA group: baseline, 106% (77–177), year 3, 95% (80–138); CGMP50, baseline, 105% (90–120), year 3, 100% (88–144); and CGMP100, baseline, 104% (85–126), year 3, 101% (87–118) (Table 2).

ANCOVA analysis adjusting for age, gender, energy intake (Kcal/d) and EAR showed that the difference in three-year EAR between AA and CGMP50 was not statistically significant (*p* = 0.717) and neither was the difference between AA and CGMP100 (*p* = 0.673). Further longitudinal analysis showed no significant differences between the groups over the three years for energy intake. 

Over the three-year period, the median percentage energy contribution from carbohydrate, protein and fat was not significantly different among the three groups (Table 3). 

The median percentage energy contribution from protein (including protein substitute and natural protein from food) was similar for all three groups. Protein provided a median of 15% (75 g) of the total energy intake, and natural protein intake supplied a median intake of 10–16% (8–12 g/day) of the total protein intake. A small but significant difference was noted for the contribution of protein substitute between CGMP50 (88%) and CGMP100 (85%) (*p* = 0.01) as well as between CGMP50 (88%) and AA (85%) (*p* = 0.007). Although the protein substitutes still provided >85% of the total protein intake in all groups, there were significant differences in the youngest age group (CGMP50) and in those children unable to tolerate extra phenylalanine from CGMP-AA (Table 3). 

Table 4 describes the total protein intake from all foods including fruits and vegetables that are usually allowed without restriction in a UK diet. The actual intake was higher compared to the “prescribed” or allocated phenylalanine exchanges (due to the small amounts of protein present in foods given without restriction). There was a significant increase in the actual intake between baseline and year 3 in AA (*p* = 0.002) and CGMP50 (*p* = 0.02) groups. The median protein intake from food was significantly different between and within the groups. CGMP50 had the lowest natural protein intake, with significant differences between CGMP50 and both AA and CGMP100 (Table 4).

These differences reflect protein tolerance, being lower in the youngest age group and those unable to tolerate the extra phenylalanine from CGMP-AA to meet all their protein substitute requirements.

### 3.3. Phenylalanine Control

There was a significant increase in blood phenylalanine levels between baseline and year 3 for AA (*p* = 0.02) and CGMP50 (*p* = 0.04), although all groups had median blood phenylalanine control within target (Table 5). This increase is an expected finding due to children reaching teenage years, when dietary adherence is known to deteriorate [43,44]. 

### 3.4. Anthropometric Data

For weight and BMI measurements, ANCOVA was applied adjusting for age, gender, weight and BMI over the three-year duration (Table 6).

#### Weight and BMI Z Scores

There was no statistical difference in three-year weight Z scores between AA and CGMP50 (*p* = 0.7), AA and CGMP100 (*p* = 0.7) and CGMP50 and CGMP100 (*p* = 0.95). Similarly, for BMI Z scores there was no statistical difference in the three-year BMI Z scores between AA and CGMP50 (*p* = 0.784), AA and CGMP100 (*p* = 0.553) and CGMP50 and CGMP100 (*p* = 407).

Using the World Health organization (WHO) definition of obesity (BMI equivalent to two standard deviations over the reference median), obesity rates between years 1 and 3 remained unchanged in the AA group (26%, *n* = 5/19), increased in the CGMP50 group from 0% to 19% (*n* = 3/16) and remained at 0% in the CGMP100 group over the three years. Overweight, defined as one standard deviation over the reference median, decreased in the AA group from 37% (*n* = 7/19) to 26%, (*n* = 5/19), remained unchanged in the CGMP50 group at 19% (*n* = 3/16) and increased in the CGMP100 group from 15% (*n* = 2/13) to 46% (*n* = 6/13). 

## 4. Discussion

This three-year longitudinal study in PKU systematically reviewed the macronutrient intake and anthropometry of children taking AA compared to CGMP-AA, with CGMP-AA provided at two different CGMP concentrations. Although satiety was not directly measured through satiety visual analog scales and hormone concentrations, the hypothesis that CGMP would enhance satiety leading to lower energy intake and slower weight gain was not observed. EAR decreased in all three groups over the three years, the most consistent decrease was in the AA group, suggesting the relationship between satiety and CGMP-AA was less convincing. Adjusting for age and gender, no differences in weight or BMI were apparent between the AA and CGMP-AA groups. Although there was no obesity present in the CGMP100 group over the three years of study, the rate of overweight increased from 15% to 45%. Our data suggest that the use of CGMP50 or CGMP100 did not reduce energy intake and thereby appetite. 

Protein substitutes based on 60 g of protein equivalent from CGMP-AA provided a typical intake of approximately 45 g/day of CGMP. We cannot evaluate from this study if this quantity may have had a significant impact on hormones such as ghrelin and GLP-1, which are important in controlling satiety. It has been shown that whey protein given to healthy volunteers at 10% of the energy intake suppressed subjective hunger but made no impact on actual energy intake. At a higher protein intake of 25% of energy intake, it increased insulin, and active GLP-1 and incretin hormones were higher, but made no difference to overall satiety and energy intake [5]. In non-PKU human studies, consensus is strong that CGMP has no effect on food intake or satiety [16,45], and weight loss after long-term consumption of CGMP has not been demonstrated [17]. Overall, subjective feelings of appetite indicate that CGMP is not critical for whey-induced satiety or energy intake. In our study, natural protein intake was vegetable based, which remained consistent over the study period for each of the three groups and was not a consideration influencing satiety.

Indirectly, CGMP may influence satiety by its prebiotic properties but a direct relationship on appetite has not been studied [46]. Microbiota play a key role in regulating energy balance and satiety, by the interaction of gut hormones and pathways such as leptin-melanocortin, a critical system controlling appetite and energy balance [47,48]. CGMP can change microbiota. In mice fed CGMP, there was a reduction in Desulfovibrio bacteria, an increase in short chain fatty acids and reduced inflammatory markers [9]. Three products, lactose, commercial CGMP (70% GMP, <2% lactose) and semi-purified CGMP (51% GMP, 4% lactose), were tested on fecal high and low diversity microbiota from healthy and elderly subjects. Both CGMP products resulted in a healthy microbiota, being more pronounced in the lower lactose CGMP preparation [49]. How this translates to satiety remains unanswered.

Only one study has examined the effect of CGMP-AA and satiety in subjects with PKU. MacLeod et al. [50] gave 11 subjects (eight adults and three children) a standard breakfast with either CGMP-AA or phenylalanine-free AA. Plasma insulin, ghrelin and amino acids were measured 180 min after breakfast. Postprandial ghrelin was significantly lowered associated with fullness in the CGMP group, with total plasma amino acids and insulin concentrations only just reaching significance between the groups. This short four-day study only measured one incretin hormone at one time point. Studies on satiety ideally need to employ a wide variation in time intervals, capturing differences in appetite and plasma amino acid profiles [51]. The carbohydrate intake was higher in the CGMP-AA breakfast, which may sustain insulin concentrations over the short study period. Results from the satiety questionnaire show 60% of the adult group were overweight. It also included only three children. Overall, these limitations render any direct interpretation of the effect of CGMP-AA on satiety challenging.

There were limitations to our study. Firstly, dietary assessments regardless of method are inaccurate, although face-to-face interviews and periodic weighed food intakes were conducted to minimize these difficulties. Satiety visual analog scales post protein substitute consumption were not conducted, although their value is subjective particularly in children and obese subjects. The AA group took different amino acid preparations but 15 of 19 consumed low energy liquid preparations. Age varied between the groups but was statistically accounted for; the influence of growth and exercise on appetite was unmeasured. No biochemical hormone markers were assessed. Both insulin and ghrelin are endocrine mediators of food intake. However, insulin is an anabolic hormone and ghrelin has growth hormone functions, further complicating any clear relationship with satiety in children who are growing and reaching adolescence. Children were not randomized to one of the three protein substitute groups; choice of group was dependent on protein substitute acceptance and blood phenylalanine control when taking CGMP-AA. One further limitation was the lack of a non-PKU control group; although dietary composition would have been different, a comparison of energy intake, weight and BMI would have been useful. Despite these limitations, no obvious impact on satiety was found between our two study groups. There are inherent difficulties in studying appetite given the behavioral and environmental factors that counterbalance the physiological regulators of appetite. 

## 5. Conclusions

In this three-year longitudinal study in children with PKU, CGMP-AA when compared to AA did not appear to influence energy intake, weight gain or BMI and by implication satiety. In PKU, there is little understanding of the optimal dietary composition needed to control appetite preventing overweight and obesity. All macronutrients have unique physiological properties that influence metabolic pathways. The impact of CGMP-AA on satiety, particularly the amounts, timing of ingestion and its effect when combined with other foods related to satiety signals remains to be fully explored.

## Figures and Tables

**Table 1 nutrients-12-02704-t001:** The nutrient composition of CGMP-AA compared with conventional AA.

Protein Substitute		CGMP-AA	Phe-Free AA *
Nutrients	Units	Per 20 g PE	Per 20 g PE
Calories	Kcal	120	124
Protein equivalent	g	20	20
Total Carbohydrate	g	6.5	9.4
Sugars	g	2.2	7.8
Total Fat	g	1.5	0.7
Docosahexaenoic acid	mg	84	134
Salt	g	0.53	0.43
Vitamin A	µg RE	283	278
Vitamin D	µg	4.5	10
Vitamin E	mg αTE	6.5	5.2
Vitamin C	mg	38	36
Vitamin K	µg	35	34
Thiamine	mg	0.68	0.70
Riboflavin	mg	0.78	0.77
Niacin	mg	8.4	8.4
Vitamin B6	mg	1.0	0.9
Folic Acid	µg	136	134
Vitamin B12	µg	1.6	1.6
Biotin	µg	63.9	63
Pantothenic acid	mg	2.7	2.6
Choline	mg	204	201
Sodium	mmol	9.0	7.3
Potassium	mmol	6.8	7.9
Chloride	mmol	0.2	3.9
Calcium	mg	407	400
Phosphorus	mmol	12	11.4
Magnesium	mg	128	125
Iron	mg	7.3	7.3
Copper	µg	748	730
Zinc	mg	7.3	7.3
Manganese	mg	1.1	1
Iodine	µg	85.7	84
Molybdenum	µg	49	48
Selenium	µg	29.9	29
Chromium	µg	29.9	29
**Amino acids**				
L-Alanine	g	0.78	0.92
L-Arginine	g	0.95	1.5
L-Aspartic Acid	g	1.12	2.37
L-Cystine	g	0.01	0.61
L-Glutamine ^1^	g	2.57	-
Glycine	g	1.2	2.35
L-Histidine	g	0.7	0.92
L-Isoleucine	g	1.35	1.62
L-Leucine	g	3.00	2.54
L-Lysine	g	0.80	1.67
L-Methionine	g	0.28	0.45
L-Phenylalanine	g	0.03	0
L-Proline	g	1.52	1.69
L-Serine	g	0.96	1.04
L-Threonine	g	2.20	1.62
L-Tryptophan	g	0.40	0.5
L-Tyrosine	g	2.24	2.38
L-Valine	g	1.09	1.86

CGMP-AA, casein glycomacropeptide; AA, phenylalanine-free amino acid; PE, protein equivalent; ***** based on liquid Phe-free amino acids (Vitaflo International Ltd); ^1^ glutamine content varies according to if the AA protein substitute is a liquid or powder.

**Table 2 nutrients-12-02704-t002:** Median daily energy intake (range) and %EAR (range) for AA, CGMP50 and CGMP100 from baseline to year 3.

	Median Energy (Kcal/Day)			Median % EAR		
Year	AA (Range) *n* = 19	CGMP50 (Range) *n* = 16	CGMP100 (Range) *n* = 13	AA (Range) *n* = 19	CGMP50 (Range) *n* = 16	CGMP100 (Range) *n* = 13
Baseline	1950 (1138–2999)	1701 (1466–2494)	1831 (1612–2591)	106 (77–177)	105 (90–120)	104 (85–126)
1	1957 (1151–3191)	1793 (1560–2681)	1917 (1642–2708)	102 (77–140)	99 (85–132)	96 (73–124)
2	1976 (1517–2669)	1828 (1512–2814)	1965 (1258–2810)	99 (72–130)	105 (90–158)	107 (85–152)
3	2120 (1111–3387)	1966 (1523–2405)	2064 (1672–3144)	95 (80–138)	100 (88–144)	101 (87–118)

AA, amino acid; CGMP, casein glycomacropeptide; EAR, estimated average requirement; CGMP50, patients taking a combination of CGMP-AA and AA; CGMP100, patients taking all their protein substitute from CGMP-AA.

**Table 3 nutrients-12-02704-t003:** Median three-year percentage (range) energy contribution for protein, carbohydrate and fat from food and protein substitute in AA, CGMP50 and CGMP100.

	Median % Energy from Protein			Median % Energy from Carbohydrate			Median % Energy from Fat		
	AA *n* = 19	CGMP50 *n* = 16	CGMP100 *n* = 13	AA *n* = 19	CGMP50 *n* = 16	CGMP100 *n* = 13	AA *n* = 19	CGMP50 *n* = 16	CGMP100 *n* = 13
Baseline (Range)	15 (10–30)	15 (9–26)	15 (11–26)	57 (45–68)	56 (43–70)	57 (45–69)	26 (15–40)	26 (17–37)	28 (17–33)
Year 1–3 (Range)	15 (9–27)	15* (10–22)	16* (12–23)	58 (42–70)	57 (46–67)	58 (48–67)	27 (16–39)	26 (18–37)	27 (18–35)

* *p* = 0.02. AA, amino acid; CGMP, casein glycomacropeptide; CGMP50, patients taking a combination of CGMP-AA and AA; CGMP100, patients taking all their protein substitute from CGMP-AA.

**Table 4 nutrients-12-02704-t004:** Median natural protein intake (g/day) (range) from food sources only and median percentage intake from protein equivalent from substitute from baseline to year 3 in AA, CGMP50 and CGMP100.

Natural Protein Intake from Food (g/d)				
**Year**	**AA** ***n* = 19**	**CGMP50** ***n* = 16**	**CGMP100** ***n* = 13**	***p* Value**
**Baseline** **(Range)**	11.5 ^*^ (5–32)	8 ^*,§^ (4–23)	12 ^§^ (5–36)	^*,§^*p* = 0.001
**1** **(Range)**	12 ^*^ (5–33)	8 ^*,§^ (4–24)	12 ^§^ (5–34)	^*^*p* = 0.0001, ^§^ *p* = 0.001
**2** **(Range)**	13 ^*^ (5–33)	9 ^*,§^ (3–23)	10 ^§^ (5–44)	^*^*p* = 0.014, ^§^ *p* = 0.02
**3** **(Range)**	12 ^*^ (6–46)	9 ^*,§^ (3–24)	13 ^§^ (6–36)	^*^ <0.0001, ^§^ *p* = 0.0006
**Median % protein intake from protein substitute equivalent (range)**				
**Year**	**AA**	**CGMP50**	**CGMP100**	***p* Value**
**Baseline** **(Range)**	85 (68–92)	87 (72–93)	85 (58–92)	
**1** **(Range)**	86 (72–92)	88 (71–93)	83 (65–91)	
**2** **(Range)**	86 (65–93)	87 (70–91)	86 (59–93)	
**3** **(Range)**	84 (68–90)	88 (75–93)	84 (68–91)	
**Year 1–3** **(Range)**	85 ^*^ (65–93)	88 ^*,§^ (70–93)	85 ^§^ (59–93)	^*^*p* = 0.01, ^§^ *p* = 0.007

AA, amino acid; CGMP, casein glycomacropeptide; CGMP50, patients taking a combination of CGMP-AA and AA; CGMP100, patients taking all their protein substitute from ^*^ significant difference between AA and CGMP50; ^§^ significant difference between CGMP50 and CGMP 100.

**Table 5 nutrients-12-02704-t005:** Median (range) blood phenylalanine concentrations from baseline to year 3 in AA, CGMP50 and CGMP100.

	Baseline Median Phe μmol/L			Year 3 Median Phe μmol/L		
	AA μmol/L *n* = 19	CGMP50 μmol/L *n* = 16	CGMP100 μmol/L *n* = 13	AA μmol/L *n* = 19	CGMP50 μmol/L *n* = 16	CGMP100 μmol/L *n* = 13
**Phe** **(Range)**	315^*^ (140–600)	255 ^§^ (170–360)	290 (200–710)	360 ^*^ (210–830)	290 ^§^ (220–430)	320 (250–895)

^*^*p* = 0.02, ^§^
*p* = 0.04. AA, amino acid; CGMP, casein glycomacropeptide; CGMP50, patients taking a combination of CGMP-AA and AA; CGMP100, patients taking all their protein substitute from CGMP-AA.

**Table 6 nutrients-12-02704-t006:** Change in weight and BMI Z scores from baseline to year 3 in the AA, CGMP50 and CGMP100 groups applying ANCOVA adjusting for variables in age and gender differences.

BMI Z Score	Baseline	36 m	Differences
**AA**	−0.15	0.63	0.3
**CGMP50**	0.17	0.85	0.5
**CGMP100**	−0.11	0.95	0.6
**Wt Z score**	**Baseline**	**36 m**	**Differences**
**AA**	−0.25	0.74	0.2
**CGMP50**	0.28	0.91	0.2
**CGMP100**	0.02	0.97	0.4

BMI: body mass index; Wt, weight; AA, amino acid; CGMP, casein glycomacropeptide; CGMP50, patients taking a combination of CGMP-AA and AA; CGMP100, patients taking all their protein substitute from CGMP-AA.

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
