# Peer review of "The Impact of the Use of Glycomacropeptide on Satiety and Dietary Intake in Phenylketonuria"

_nutrients, 2020, doi:10.3390/nu12092704_

Round 1

Reviewer 1 Report

The use of glucomacropeptide allows to know the effect of satiety in patients with phenylketonuria.

This paper adds the conclusions to three previous published researchs

The paper is well written and easy to understand

The conclusions are consistent with the evidence presented and they address the main question posed by the authors 

Author Response

Dear Reviewers

Thank you for your comments which have been helpful and constructive.

We have replied to these and made amendments in red in the manuscript.

The reply is attached.

Thank you Anne Daly

Reviewer 2 Report

General Points

Overall this is a competent if negative results study on the possible effect of different PKU-treatment diets on satiety in children suffering from the condition. I agree with you main conclusions based on the evidence you submit. Most of my request edits are minor revisions. However, on the anthropometric data you must report more comprehensively than you have currently (see comments below). In addition, I have two major questions that I would like an answer to but do not require any changes to the paper, other than the authors might see fit to clarify:

  1. I can understand why it is difficult in longitudinal nutritional research on human subjects to have true control groups (in this case a health cohort on the same diet as the PKU subjects). I also understand you are only interested on comparing within PKU patients. But I would suggest you modify you conclusion statements to incorporate some indication about this limitation.
  2. Why did you not (as you state yourself) run a standard battery of biochemistry test to measure satiety changes directly? i.e. the visual analogue scale and endocrine plasma tests? It seems an obvious point and given the scope of the study does not seem to have been a significant extra burden on data collection from the patients.

Specific Edits Requested and Clarifications

Introduction

  1. A major part of your introduction is to outline what is known above the link between amino acid/protein content of diet and satiety, which is an essential rationale for why you might require a study of the effects of different protein replacements in PKU sufferers. There are some problems with your introduction on the research into the effect of dietary AA/protein composition on satiety, address the following:

a) Reference 21, 22 are not correct. These papers do not provide direct evidence linking the plasma concentrations of these specific AAs and satiety. Please find specific primary research that provides such evidence. 

b) Refs 25, 26 similarly, are not adequate. One (26) is very old and your own review (ref 25) provides far more comprehensive up-to-date information. Please provide primary research citations that provide the evidence for the assertion made in this sentence i.e. directly correlating increased blood plasma AA concentrations to satiety.

c) In the sentence 'equally, it is well established that amino acids do not undergo digestion...' This is a nonsensical statement. Of course amino acids undergo 'digestion'. what you mean is that they are not chemically catabolised in the gut by enzymatic or acidic action. Please correct. Just to point out, the direct absorption of AAs you refer to is quite ambiguous as well. The absorption of individual AAs and 2-3mer peptides crosses two cell membranes of the epithelial cells before entering the portal vein. The word 'rapid' here is also rather a subject term: rapid compared to what. Consider revise this sentence to make it more accurate and less ambiguous. 

d) Refs 24 and 25 are also incorrect. 25 is a review used where you should be citing primary research. 24 does not provide any specific evidence for the absorption per se of its rapidity that I can see. Can I help? The physiological basis of this claim comes from the established principle that single AAs and 2 and 3 AA long peptides are absorbed from the small intestine across the epithelial brush-border membrane. Any peptides longer than 2 or 3mers must be 'digested' via chemical hydrolysis first. As a result the central question to this claim is: how long is the CGMP peptide? If it is a 2/3 AA peptide it will be absorbed by the PEPT1 transporter of the small intestine brush-border membrane.  If it is longer it will first undergo hydrolysis by brush-border peptidase enzymes before it can be absorbed. In any case, these references do not report evidence for the specific 'rapid' absorption of CGMP. If you can find such evidence then cite it. If there is no direct study on the absorption efficiency, rate ect of cGMP than all you can do is use the principles of peptide absorption I have outlined (with citations) to state why CGMP will/will not be absorbed directly from the gut and then enzymatically broken down to single AAs before leaving the epithelial cells into the portal vein. Please fix and re-write.

e) Again Refs 11, 15, 20 are not correct. You seem to be using citations of papers from your specific field (protein nutrition and absorption, and satiety) to reference discoveries made in others. These are not the primary research citations which established any of these gut hormones as satiety signals. If you provide the information please find the correct citations. There is nothing wrong with the citations you use to reference the phenomenon of dietary-regulated satiety. But they are incorrect to cite for specific biochemical and autonomic nervous system components of this signalling.

f) The sentence 'The absorption kinetics of cGMP...has not been reported' not only contradicts your assertion that cGMP is 'rapidly' absorbed (see my note above), it also does not make sense. What does it mean for the absorption kinetics for a particular chemical nutrient to be 'adapted' to a specific medical conditions like PKU? Why would someone having PKU (which causes the inability to catabolise by mutations on Phenylalanine hydroxylase) effect the absorption kinetics of a peptide supplements designed to alleviate the conditions? None of this is explained. in any case your aim is simply to see if different PKU treatment diets effect satiety in PKU sufferers. Absorption kinetics of these peptides of AAs in general is not relevant unless you are measuring them in the results, which you do not. If you mean by 'kinetics' here the amount of these protein substitutes taken up per day then please change this sentence to reflect this. 

2. One piece of information missing from your introduction and rationalising the need for your study is what specific effects of protein/AA composition effect satiety have been discovered that might lead you to specifically suspect CGMP does effect satiety itself? Is there a direct link between Phe and satiety? This was unclear, or there is none. Please clarify. I must saw it seems a rather tenuous rationale to simple study the effect of PKU treatment diets on satiety without having some more direct indication that this may be the case. At the moment its seem like you are simply saying 'Protein is known to affect satiety, therefore we wanted to study the effect of the absence of dietary Phe (in PKU patients) on satiety.

Methods

1.Table 1: Chloride concentrations in CGMP is very low compared to the AA diet. Chloride will form the counter-ion for most of the cation species in the food (sodium, calcium etc). This CGMP chloride concentration seems very low and not to balance the cation concentrations. I can see no other anion that would have acted as the counterion in added salts. Check this concentration. Also for the The-free AA diet no Glutamine conc is given - is it really absent or is it just now known? 

2. Study design: for the non-nutrition study expert please outline the regular dietary-based treatments for PKU children as a background. This will give context to the diets, their composition and how they are used to alleviate the effects of PKU.

3. Blood Phenylalanine levels: This method requires more details from the analytical chemists who conducted to the MS/MS ID and quantification. More detail required on the protocol, the machine used, identification etc.

4. Why is the median +/- IQR used throughout the study instead of, say the mean +/- S.D? Is this a standard practice in nutritional research?

Results and Discussion

  1. Please clarify/correct the sentence 'The median phenylalanine concentration at baseline were not... (1st paragraph results). Are the values in brackets the range? what are there units? The same as the median values? Where is there error term for your measure of central tendency (median). You should always be quoting an error term - for median it should be the IQR values. For example median +/- IQR units.
  2. The previous points' comments on IQR and values in brackets need to be fixed for the while results. 
  3. Please do not cite the tables in the sub-headings (e.g. tables 2, 3 and 4). They should be cited when first mentioned in the text.
  4. If you are quoting the range of values in brackets after all the median values, then Table 2 value for Median Energy, AA column, year 3 (2120 Kcal/day, 5111-3387 I brackets) does not make sense. How could the median value lie outside the range of all values collected?
  5. Results 3.2.1: why is an ANCOVA comparing CGMP50 to CGMP100 not reported? 
  6. Results 3.2.2: The first sentence: Is the information reported in Table 3 not readily available from the manufacturer of CGMP and the AA diets? In any case they can be easily calculated. Or am I missing something and you've included the natural protein intake from food (other than the diets) in this table? If so please clarify.
  7. The sentence 'The actual protein intake compared ... without restriction in a UK diet.' does not make sense. Please re-write.
  8. As with above comment, please cite table 4 in its appropriate place in the text.
  9. Results 3.2.2: The sentence 'These differences reflect protein tolerance, being lower...' Fair enough given CGMP diet has 0.03 g of The that the AA diet does not. But is this a known previous finding? Or is it your assertion/interpretation of the protein consumption discrepancies you report? Please clarify and cite if it is a known results. If not, it should be discussed (in discussion).
  10. Your median values would all have the IQR as the +/- error term, no? If you don'e agree explain why not? 
  11. Results 3.2.3: would you expect this plasma concentration increase in Phe on both diets for PKU children over 3 years? Please clarify in the results and if this is unexpected (I suspect not) it should be discussed.
  12. Section 3.2.4: This is really the most important results of your study according to your aim. These anthropometric measures are really you only way of implying if the different diets have any effect on satiety (the rest are just controls for consistent energy and macronutrient intake). I would suggest you point this out or emphasise it in the results for the reader.
  13. Section 3.2.4: the formatting of this section looks like you are almost reporting tabulated results in the text. Please write more clearly and fully the results as true sentences. 
  14. Section 3.2.4: You need to report the raw data for Table 6 as it is the central results of your study. Please provide the age, gender, weight and raw BMI scores for all subjects. In addition, when communicating the obesity rates you quote the individual subjects (e.g. n = 5/19) so why not show the raw anthropometric data for each? 
  15. Section 3.2.4: The paragraph where you compare to the WHO obesity standards and quote as a percentage the numbers of each cohort who could be defined as obese after 3 years needs to be re-written to make it clearer. It needs one or two edits before it is passable for publication. 
  16. Discussion: first sentence. Because of your lack of reporting the raw anthropometric data (point 14) I do not agree with this sentence. 

Author Response

(The authors gave the same response as above.)
